# Evaluation of Semi-Continuous Pit Manure Recharge System Performance on Mitigation of Ammonia and Hydrogen Sulfide Emissions from a Swine Finishing Barn

**Jisoo Wi** [1] , **Seunghun Lee** [1], **Eunjong Kim** [1] , **Myeongseong Lee** [1] , **Jacek A. Koziel** [2,*]
and **Heekwon Ahn** [1,*]

[1]   Department of Animal Biosystems Sciences, Chungnam National University, Daejeon 34134, Korea;
      jswi@cnu.ac.kr (J.W.); huny9261@cnu.ac.kr (S.L.); ejkim0128@cnu.ac.kr (E.K.); leefame@cnu.ac.kr (M.L.)
[2]   Department of Agricultural and Biosystems Engineering, Iowa State University, Ames, IA 50011, USA
*   Correspondence: koziel@iastate.edu (J.A.K.); hkahn@cnu.ac.kr (H.A.);
    Tel.: +1-515-294-4206 (J.A.K.); +82-042-821-5785 (H.A.)

**Abstract:** In this research, for the first time, we present the evaluation of a semi-continuous pit manure recharge system on the mitigation of ammonia ($NH_3$) and hydrogen sulfide ($H_2S$) emissions from a swine finisher barn. The pit recharge system is practiced on many swine farms in the Republic of Korea, primarily for improving air quality in the barn. It consists of an integrated waste management system where the fraction of stored manure is pumped out ($10\times$ of the daily production of manure, $3\times$ a day); solids are separated and composted, while the aerobically treated liquid fraction is then returned to the pit. We compared emissions from two 240-pig rooms, one equipped with a pit recharge system, and the other operating a conventional slurry pit under the slatted floor. Mean reduction of $NH_3$ and $H_2S$ emissions were $49 \pm 6\%$ and $82 \pm 7\%$, respectively, over 14 days of measurements. The removal efficiency of $H_2S$ was higher than $NH_3$, likely because the pH of aerobically treated liquid manure remained slightly above 8. More work is warranted to assess the N balance in this system and the emissions of odor and greenhouse gasses (GHGs). It is also expected that it will be possible to control the $NH_3$ and $H_2S$ removal rates by controlling the nitrification level of the liquid manure in the aerobic treatment system.

**Keywords:** gas emissions; air quality; odor control; livestock production; manure management; Republic of Korea

## 1. Introduction

Gaseous and particulate matter emissions from livestock farming impact the local and regional air quality. However, gaseous emissions composition is very complex. Lo, et al. [1] reported nearly 300 volatile organic compounds (VOCs) emitted from swine manure, with a significant fraction being odorous. Besides VOCs, gaseous emissions from livestock facilities contain gases such as ammonia ($NH_3$), hydrogen sulfide ($H_2S$), and greenhouse gases (GHGs). $NH_3$ is considered an environmental pollutant because when released into the atmosphere, it can cause acid rain and soil acidification [2]. Also, the ammonia present in surface water can cause the eutrophication of rivers [3]. Ammonia can also form secondary fine particulate matter (PM) in the presence of $NO_x$ or $SO_x$. The interactions of odorous VOCs with PM are not well understood, but it is evident that PM itself can serve as a carrier of odor and that fine PM can be more odorous [4]. Researchers have discussed potential adverse effects of

air quality in the context of animal production to health, especially respiratory issues in both workers and animals [5,6]. Gaseous emissions can elicit complaints from neighbors [7,8].

Extensive measurements of odorous gas and PM emissions have been conducted to account for a wide range of site-specific conditions such as the animal species, facility size and type, manure management, climate, ventilation schemes, and others [9–14]. It is well known that manure management can have a significant impact on gaseous emissions [15]. Measurements require a major investment of resources and time to account for variability. Still, odor and gaseous measurements are confounded by measurement biases and inconsistencies [13,16–18]. Researchers agree that $NH_3$ and $H_2S$ are relatively easy to measure in real time and can (to a limited extent) serve as surrogates for the assessment of odor [19].

Various techniques for reducing odor, gaseous and PM emissions have been developed. Maurer et al. [20] reviewed the effectiveness of technologies for mitigation at the housing, manure storage and treatment and land application scales for swine, poultry, beef cattle, and dairy. The odor reduction techniques for animal housing (barns) can be divided into 'end-of-pipe' and 'source-based' methods [21]. Some of the end-of-pipe techniques include biofilters [22,23], windbreaks, wet scrubbers and ultraviolet light [24,25]. The source-based approaches can manage emissions at the odor source, usually addressing emissions from manure. Various topical additives to swine manure surface such as biochar [26], soybean peroxidase [27,28], and poultry manure such as zeolites [29], and bioactive sorbents [30] have been tested. Because swine barn odor is mainly caused by emissions from manure [31], the source-based methods are aimed at proper manure management. Source-based methods comprise feed management, stocking density reduction, and the cooling of manure [21]. Feed management can be expected to reduce $NH_3$ emissions by minimizing N excretion by controlling the N content in feed or phase feeding [3,32]. Also, reducing the stocking rate can reduce farm productivity and profitability, and cooling the manure requires a cost. Even with the proper mitigation of gaseous emissions at the source or storage, land application of manure (that is a desired nutrient cycling practice) can also be a source of emissions [33].

The manure pit recharge system could be considered as one of the source-based approaches for manure management. In this method, the pit is periodically recharged with a liquid. Although fresh water can be used as recharge liquid [34], recycled effluent from an anaerobic lagoon is more practical [35,36]. It is known as a manure collection system and is also reported to be an effective method to prevent the volatilization of odorous substances by diluting the swine manure. Kai et al. [34] reported the $NH_3$ emission reduction rate for a pit recharge system with fresh water ranging between 54 and 88% when compared with previously reported gas emission from conventional manure collection systems. Similarly Lim et al. [35] reported the reduction rates of $NH_3$ and $H_2S$ were 52–63% and 17–41%, respectively when recharging frequencies ranged from 7 to 14 days with lagoon effluent. In addition, Blunden et al. [36] reported emission rates for $NH_3$ and $H_2S$, and Ha and Kim [37] reported $NH_3$ and $H_2S$ concentrations inside the swine barn equipped with a pit recharge system without explicitly reporting the reduction of gaseous emissions. The summary of the literature on the performance of pit recharge systems is shown in Table 1.

To date, the pit recharge system described in literature used either water [34] or treated manure [35,36] where the entire shallow manure pit is drained (a batch mode) and recharged with a treated lagoon effluent. The use of water can be problematic as it is an expensive resource in some livestock production areas, and it can also increase the volume of manure. The batch recharge, while feasible, adds to the personnel management workload and the need for additional scheduled monitoring and operation (sometimes not feasible in highly automated production systems).

**Table 1.** Summary of $NH_3$, $H_2S$ and odor emission from swine barns equipped with a batch-mode pit recharge system reported in previous studies.

| Study | Experimental Scale & Location | Growth Stage & Size | Description of Barn | Recharged Liquid & Volume | Recharging Frequency | Experimental Period and Season | Gas & Odor; Emission Rate $(g\ d^{-1}AU^{-1})$ [1] | Measured Gas Emissions Reduction Rate (%) |
|---|---|---|---|---|---|---|---|---|
| Lim et al. (2004) [35] | Research barn Indiana, US | 78–97 kg finisher, 75 pigs | Shallow pit (1.1 m deep), Fully slatted | Anaerobic treated lagoon effluent, 72 L head$^{-1}$ | 1 wk, Batch mode | May–July (3 wks) | $NH_3$; 10 $H_2S$; 0.16 Odor; 2.6 $OU_E\ s^{-1}$ | 63 41 |
| | | | | | 2 wks, Batch mode | May–July (6 wks) | $NH_3$; 12 $H_2S$; 0.34 | 52 17 |
| | | | | | 6 wks, Batch mode | March–May (7 wks) | $NH_3$; 11 $H_2S$; 1.42 Odor; 25 $OU_E\ s^{-1}$ | Not reported [2] |
| Kai et al. (2006) [34] | Research barn Denmark | 25–45 kg grower 20 pigs | Shallow pit (0.3 m deep), Fully slatted | Fresh water, 6 L head$^{-1}$ | 1 wk, Batch mode | 1 wk | $NH_3$; 17–23 Odor; 6.0 $OU\ s^{-1}$ | Not reported |
| Blunden et al. (2008) [36] | Commercial barn North Carolina, US | 38–88 kg finisher 842–896 pigs | Shallow pit, Fully slatted | Anaerobic treated Lagoon fluid (No information about volume) | 1 wk, Batch mode | February (6 days) | $NH_3$; 40.8, $H_2S$; 4.2 | Not reported |
| | | | | | | April (8 days) | $NH_3$; 37.1, $H_2S$; 3.3 | |
| | | | | | | June (6 days) | $NH_3$; 29.5, $H_2S$; 1.2 | |
| | | | | | | October (6 days) | $NH_3$; 14.3, $H_2S$; 1.7 | |
| Ha and Kim (2015) [3] [37] | Commercial barn Republic of Korea | Not reported | Not reported | Aerobically treated liquid manure | Not reported | Not reported | Not reported | Not reported |

[1] 500 kg of live animal weight; [2] Only absolute emissions are reported; [3] Ha and Kim (2015) measured concentration of $NH_3$ and $H_2S$, not emission rates.

Thus, approximately 100 swine farmers in the Republic of Korea have adopted a more frequent semi-continuous pit recharge system. Recharging manure is diverted daily from the aerobic treatment of swine manure that is part of a comprehensive manure treatment that aims at reducing odorous emissions while generating high-quality liquid fertilizers. Still, very little is known about the actual performance of this kind of highly-integrated pit recharge system on $NH_3$ and $H_2S$ emissions from a swine barn. Ha and Kim [37] measured gas concentrations from a barn with conventional slurry pit and three barns equipped with a pit recharge system. Reduction of $NH_3$ and $H_2S$ concentrations were reported as 35–84% and 0–100%, respectively, but emission rates have not been reported.

In this research, for the first time, we tested the performance of the semi-continuous pit recharge system on mitigation of $NH_3$ and $H_2S$ emissions from a swine finisher barn in Korea. The pit recharge system using aerobically treated liquid manure (treated manure diverted back into the pit) has been developed for gaseous emissions reduction method for swine barn (Figure 1). This method (of daily removal of manure) can improve the air quality inside a barn by reducing odorous gas generation [38]. Therefore, the number of farms using a pit recharge system has been steadily increasing in the Republic of Korea, but no studies reported the efficiency of $NH_3$ and $H_2S$ emission reduction from swine barns equipped with a pit recharge system with aerobically treated manure.

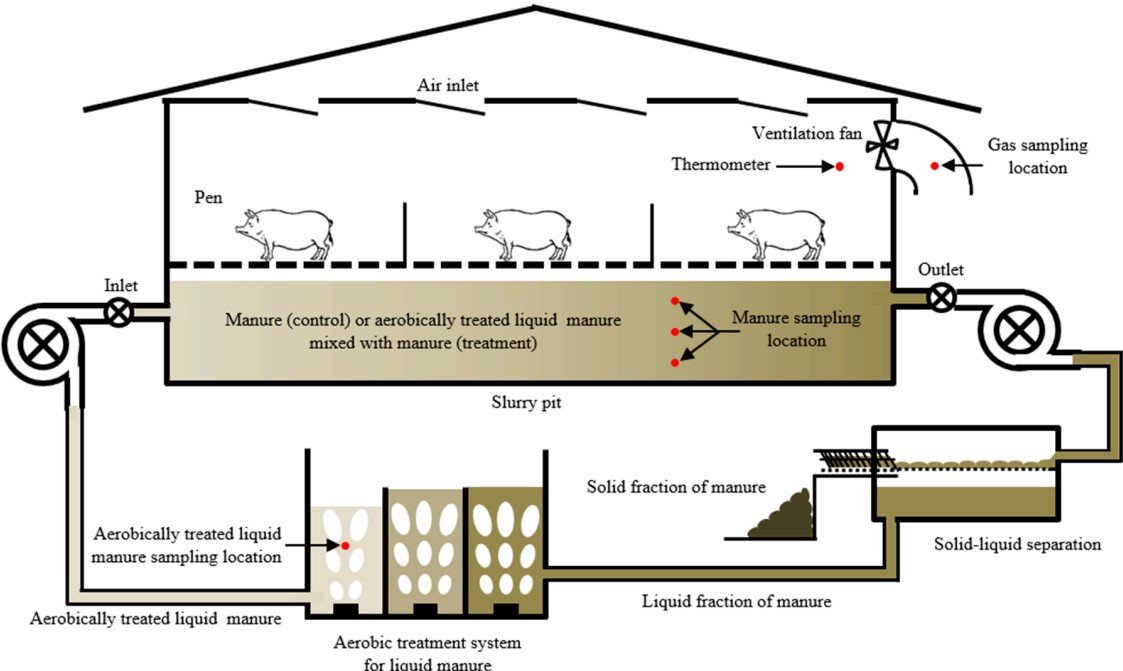

**Figure 1.** Pit recharge system with aerobically treated liquid manure. After solids separation, the liquid fraction goes to the sequential aerobic treatment system, which can be considered as equivalent to autothermal thermophilic aerobic digestion (ATAD) in Europe [39]. The aerobically treated liquid manure is used as the recharging liquid of the pit recharge system.

The main objective of this study was to evaluate the performance of a pit recharge system on $NH_3$ and $H_2S$ emissions from a swine finishing barn for a typical farm operation in the Republic of Korea. Specifically, we compared $NH_3$ and $H_2S$ emissions between conventional a slurry pit and a pit recharged with aerobically treated manure.

## 2. Experiments

### 2.1. Description of Farm and Design of Experiment

The experimental site was a commercial pork production farm, located in Buyeo, Chungnam Province in the Republic of Korea. The experiment was carried out for two weeks from 30 September

to 13 October 2017. During this period, the average temperature outside the barn was 17.3 ± 2.9 °C. Two identical swine rooms (with and without a pit recharge system) in one confinement building for finishing pigs were used to evaluate the effect of a pit recharge system to gas emission rates (Figure 1, Figure A1). Each tested room was representative of a total of three rooms operating in the same mode. A room with conventional slurry pit was used as a control and the other one was pit recharged with a total 12 m$^3$ (approximately 10 times the daily production of manure from 240 pigs; designed based on the assumption of 5.1 kg of combined manure and waste water produced daily per head) of aerobically treated liquid manure three times (6 A.M., 12 P.M., 6 P.M.) daily. The total amount of recharged aerobically treated liquid manure in a day was about 7.3% of the total stored manure in the pit. Manure and recharged liquid was filled to about 70% of the total pit volume and kept at a constant depth (82 cm) throughout the experiment period. The manure from rooms 5, 6 and 7 (Figure A1) was treated collectively, i.e., each room was pumped out automatically on a round-robin basis. Thus, the recharging liquid from the aeration system was a mixture of treated manure from three pits (rooms 5 to 7). Manure from the conventional slurry pit is pumped out every 2–3 months and transported to a centralized manure treatment plant. The manure in the slurry pit was totally emptied when the pigs were introduced into the control room first, then the amount of manure slowly increased while the pigs growing. The manure volume of control room was about 70% of total pit volume (79 cm depth at day 13), almost the same depth to the pit recharged room, when the gas emissions were monitored (Figure A2). The treated liquid manure from pit recharge system is considered as a "liquid fertilizer" and is periodically pumped out to a large on-farm storage tank, and then land applied during the growing season (e.g., 1–3 times per year for rice). Each room had 240 pigs weighing approximately 80 kg, fed with feed containing 17.5% crude protein (Table 2). The stocking density was 0.79 m$^2$ head$^{-1}$. The rooms had a fully-slatted floor. Manure pits (1.2 m deep in each room) were separated and managed independently in the conventional storage system (Figure A2). Fixed walls separated pits under each room, i.e., there was no air exchange between rooms or pit headspaces. Air ventilation system was separated in each room and managed independently. The eight air inlets (one-side baffle type, 300 × 300 mm) were in the ceiling, and three exhaust ventilation fans were wall-mounted (Figure A3). One primary ventilation fan (Φ 550 mm) operated continuously at a constant rate (88 m$^3$ min$^{-1}$), and the others (Φ 1000 mm) operated with variable speed, 110~210 m$^3$ min$^{-1}$, to maintain set point room temperature (25 °C). The operation of both fans is mostly in the summer season, and during the whole experimental periods for this study, only one of Φ 1000 mm fans had operated. The gas sampling location of each room was immediately downstream of each continuously operated fan.

**Table 2.** Characteristics of feed used in this study.

| Item | Contents (%, d.b. [1]) |
|---|---|
| Crude protein | 17.48 |
| Crude fiber | 5.16 |
| Fat | 2.91 |

[1] dry basis.

### 2.2. Ammonia and Hydrogen Sulfide

The real-time monitoring system (OMS-200, Smart Control & Sensing Inc., Daejeon, Rep. of Korea; Figure A1) equipped with electrochemical gas sensors of Membrapor Co. (Wallisellen, Switzerland) were used to measure NH$_3$ (NH3/CR-50) and H$_2$S (H2S/C-50) concentrations. Both gas sensors were calibrated with standard gases on the day before the start of the experiment. The detailed performance data of both sensors are shown in Table A1. The real-time monitoring system sampled room air continuously at 2 L min$^{-1}$ for 5 min which was followed by flushing with ambient air for 10 min. This was done to minimize the risk of sensor overload and signal drift.

Airflow measurement assembly (AMA, MJ Tech., Seoul, Rep. of Korea) was installed in each exhaust fan (Figure A1) and calibrated with hot wire anemometers at varying ventilation stages. The calibrated AMA provided a real-time ventilation rate to the OMS-200. The status of ventilation fan operation was monitored using AC-to-DC power supply adapters which provide fan power voltages and on/off signals to the monitoring system. Room temperature was measured with thermometer with integral PTAT (proportional to absolute temperature) silicon transistor (Econarae, MHTP-485S). Finally, the $NH_3$ and $H_2S$ emissions from each tested swine room were estimated with the following equation:

$$\text{Gas emission per hour (g h}^{-1}\text{ head}^{-1}) = (C_E \times V \times \frac{273.15 \times \text{MW}}{(273.15 + T_E) \times 22.4 \times 10^3} -$$
$$C_A \times V \times \frac{273.15 \times \text{MW}}{(273.15 + T_A) \times 22.4 \times 10^3}) \times 60 \div heads \tag{1}$$

$C_E$ = Gas concentration of exhausted air (mL m$^{-3}$)
V = Room ventilation rate (m$^3$ min$^{-1}$)
MW = Molecular weight of target gas (g mol$^{-1}$)
$T_E$ = Exhaust air temperature (°C)
$C_A$ = Gas concentration of ambient air (mL m$^{-3}$)
$T_A$ = Ambient air temperature (°C)

*2.3. Manure Analysis*

The recharging manure (aerobically treated) was sampled at the first (day 0) and final (day 13) days of the experiment (Figure 1), and recharged liquid mixed with manure was collected from the slurry pit under swine room once (day 13) (Figure A3). Manure from the conventional slurry pit (control) was sampled on day 13 (Figure A3). Samples from each pit were collected at three different heights of 60 cm (shallow), 40 cm (middle), and 20 cm (deep) from the bottom of the pit (Figure A2). Manure samples were stored below 4 °C and analyzed for total solids (TS), volatile solids (VS), pH, electric conductivity (EC), total nitrogen (TN), and ammonium nitrogen ($NH_4$-N). TS and VS were analyzed with the standard American Public Health Association (APHA) methods. The pH and EC measurements were conducted using a digital pH meter with a combination glass electrode (Thermo Scientific, Orion 4 Star pH, and EC conductivity benchtop meter). The TN content in manure was measured using the modified Gunning method (utilizing sulfuric-salicylic acid mixture, a.k.a. total Kjeldahl nitrogen). To detect $NH_4$-N in manure, the photometric analysis was used (Thermo Scientific, Gallery Discrete Analyzer).

*2.4. Statistical Analysis*

We analyzed the Durbin-Watson value of each dataset with SPSS Statistics (IBM Corp., version 24) and judged whether the data were suitable for the *T*-test. The prescreened data were then evaluated with Origin Pro software (Origin Lab, version 9) for statistical significance using two-sample *T*-test. A significant difference between control and treatment was determined at a significance level of $p < 0.05$.

## 3. Results

*3.1. Ammonia Emissions of Each Room*

Figure 2a illustrates the measured $NH_3$ concentrations at each primary fan in control (conventional slurry pit) and treatment (pit recharge system) room during the entire experimental period. In both swine room, $NH_3$ concentration decreased through whole periods, and within treatment, the fluctuations of concentration had a distinct diurnal pattern. The ranges of $NH_3$ concentrations in control and treatment were 7.3–35.5 ppmv and 3.7–25.0 ppmv, respectively.

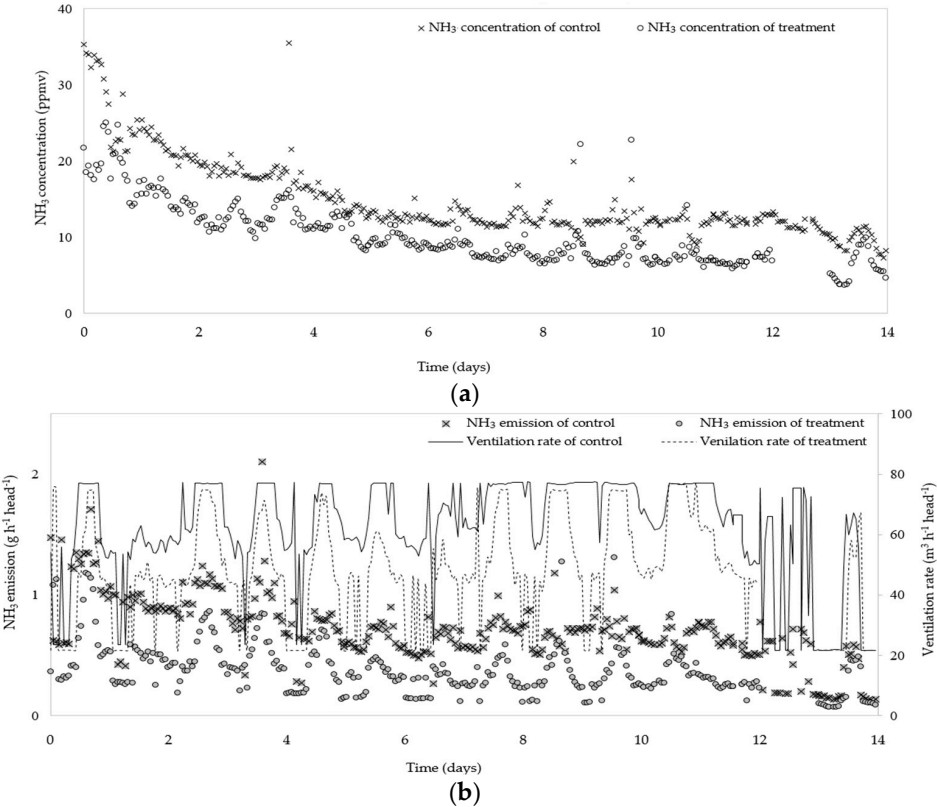

**Figure 2.** Comparison of measured $NH_3$ concentrations and emissions from control (conventional slurry pit) and treatment (pit recharge system). (**a**) Measured concentration; (**b**) estimated emission and ventilation rates.

The ranges of ventilation rate of both rooms were 22–77 m$^3$ h$^{-1}$ head$^{-1}$. Due to the primary fan operating with constant rates (88 m$^3$ min$^{-1}$), and the secondary fan operating at 110–210 m$^3$ min$^{-1}$, there was a gap between 22 m$^3$ head$^{-1}$ h$^{-1}$ to about 45 m$^3$ head$^{-1}$ h$^{-1}$ (Figure 3). The ranges of $NH_3$ emission rate were 0.1–2.1, 0.1–1.3 g h$^{-1}$ head$^{-1}$ in the and treatment, respectively. The hourly $NH_3$ emission rates showed a correlation with the ventilation rates of both rooms (Figure 2b). Correlation coefficients were calculated to examine the relationships between ventilation rates and $NH_3$ concentrations and emissions. The correlation coefficients of ventilation rate and $NH_3$ emission were 0.61 and 0.69 in control and treatment, respectively (Figure 3). On the other hand, there was no apparent correlation between $NH_3$ concentration and ventilation rate in both rooms, i.e., the coefficients of control and treatment were −0.06, 0.07, respectively.

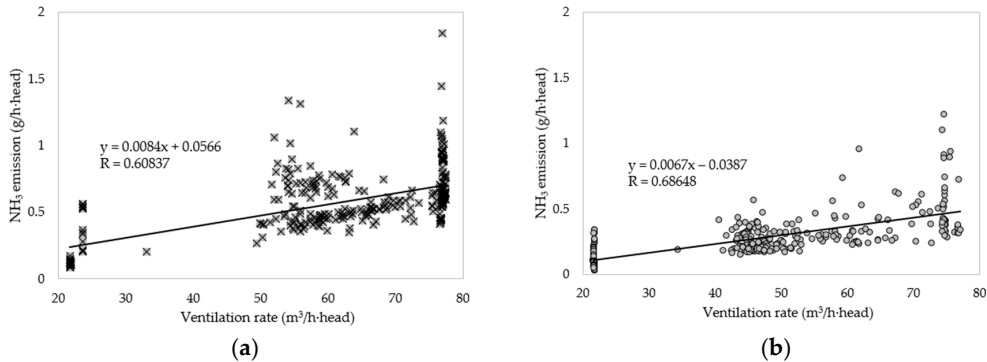

**Figure 3.** Correlation between ventilation rate and $NH_3$ emission in control (conventional slurry pit) and treatment (pit recharge system). (**a**) The correlation coefficient of ventilation rate and $NH_3$ emission in control (conventional slurry pit) was 0.61, (**b**) 0.69 in treatment (pit recharge system).

Figure 4 shows the typical diurnal variations of $NH_3$ concentrations and emissions, outside and inside temperatures, ventilation rates of control and treatment (shown for day 5 of the experiment). The room temperature was tightly controlled at around 25 °C, which was the desired set point. To maintain this set point, the ventilation rate in daytime increased with the increasing the temperature outside. The $NH_3$ concentrations were stable and ranged from 12–15 ppmv and from 8–12 ppmv at control and treatment, respectively. However, the $NH_3$ emissions reflected the diurnal ventilation rate pattern.

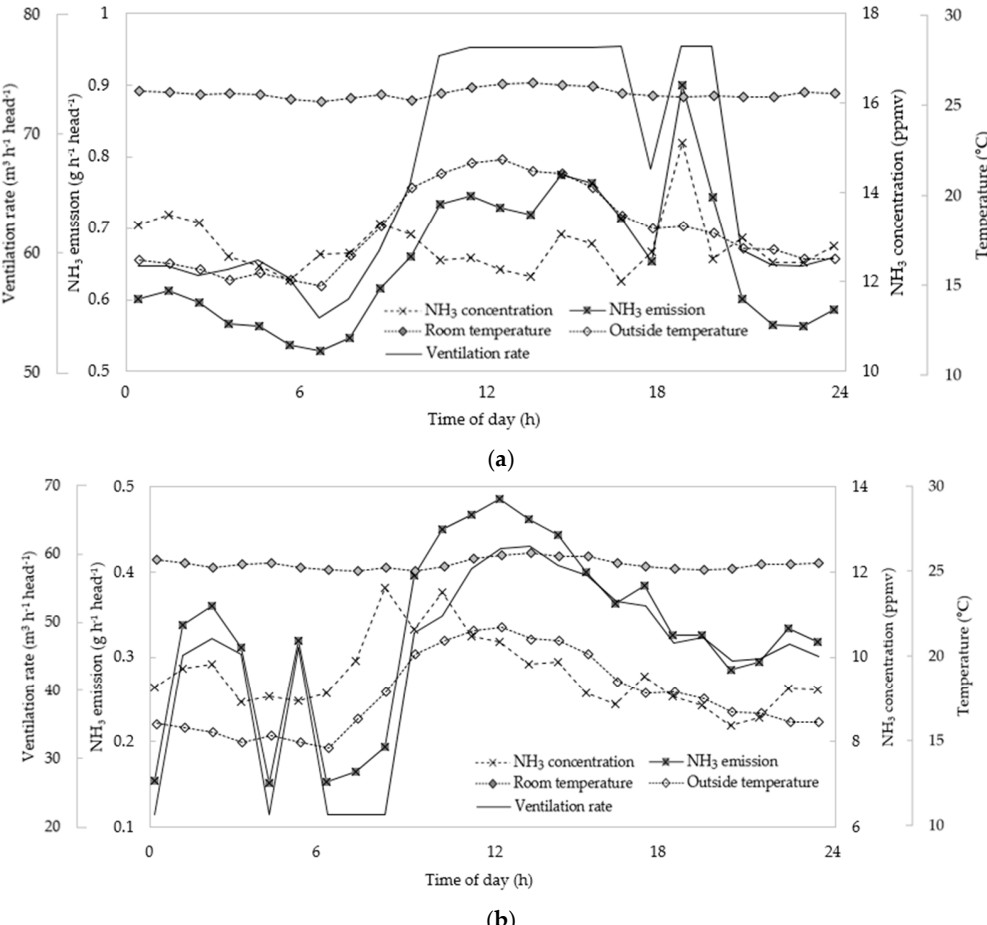

**Figure 4.** Diurnal variations of temperature, ventilation rate and generated $NH_3$ from (**a**) control (conventional slurry pit) room, (**b**) treatment (pit recharge system) room on day 5.

*3.2. Hydrogen Sulfide Emissions*

The real-time measured $H_2S$ concentrations and estimated emissions are shown in Figure 5. The large difference in $H_2S$ concentrations was observed between the control (conventional slurry pit) and treatment (pit recharge system) room. The range of $H_2S$ concentration in control was 488–2310 ppbv, while 84–1378 ppbv measured in treatment. From day 12 of the experiment, $H_2S$ concentration in control increased because the ventilation rate decreased to maintain room temperature. As with $NH_3$, the emissions of $H_2S$ reflected the ventilation rates of each room.

Figure 6 presents the $H_2S$ concentration plotted against the ventilation rate in control and treatment rooms. At the control, $H_2S$ concentration and ventilation rates were negatively correlated, the coefficient was −0.58. It is likely caused by a dilution effect due to the increased ventilation rate. In the treatment, the correlation between $H_2S$ concentration and ventilation rate was poor (R = 0.01).

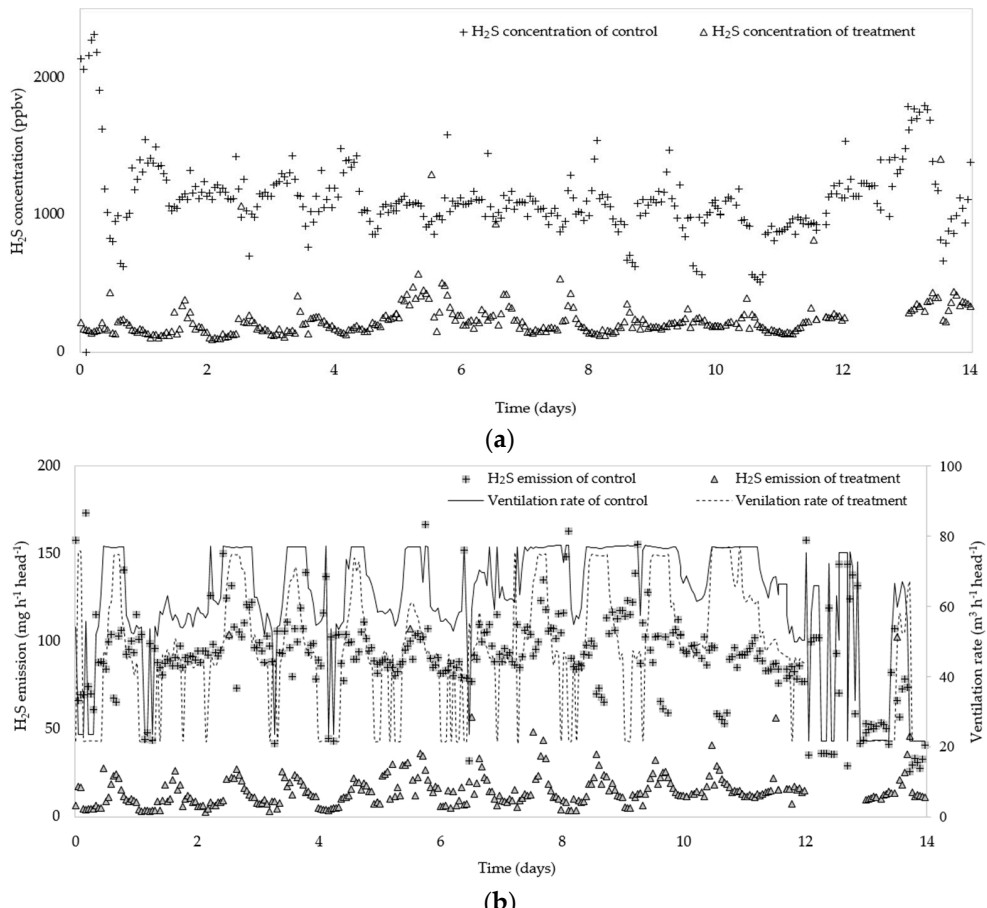

**Figure 5.** Comparison of measured H$_2$S concentrations and emissions from control (conventional slurry pit) and treatment (pit recharge system). (**a**) Measured concentration (ppmv); (**b**) estimated emission and ventilation rate.

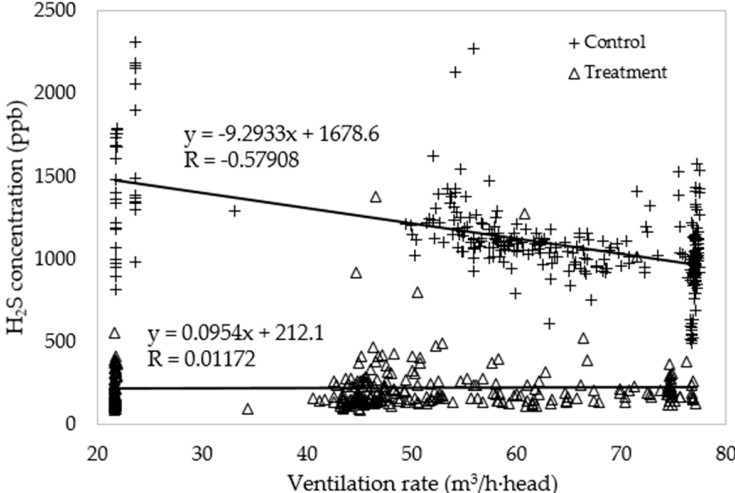

**Figure 6.** Correlation between ventilation rate and H$_2$S concentration in control (conventional slurry pit) and treatment (pit recharge system).

The H$_2$S emissions and ventilation rates indicate correlation in both rooms where the correlation coefficients were 0.69, 0.44 in the control and treatment, respectively (Figure 7).

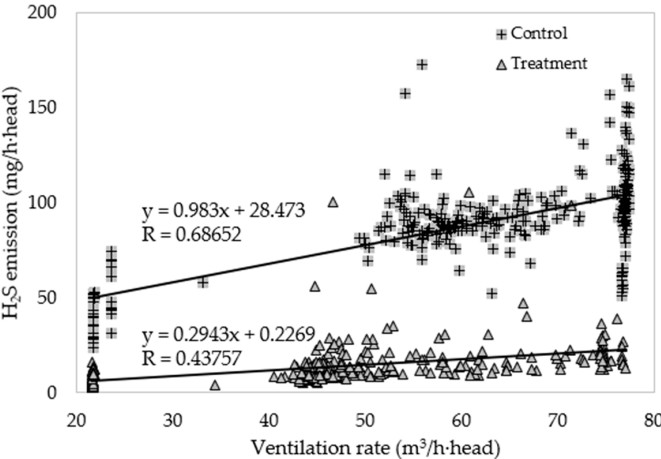

**Figure 7.** Correlation between ventilation rate and $H_2S$ emission in control (conventional slurry pit) and treatment (pit recharge system).

Figure 8 shows typical diurnal variations of $H_2S$ concentration and emission, outside and inside temperature, a ventilation rate of control and treatment (shown for day 5 of the experiment). In control (Figure 8a), the $H_2S$ concentration peaked at around 18:00, and the $NH_3$ concentration was also peaked at the same time (Figure 4a). This is presumably due to random operations on the farm; i.e., workers needed to enter rooms for maintenance, surveys, repairs of feed bin, at that time. In treatment, the $H_2S$ concentration and emission during a day seemed to be less affected by the variation of temperature and ventilation rates, and the highest concentration was measured around noon.

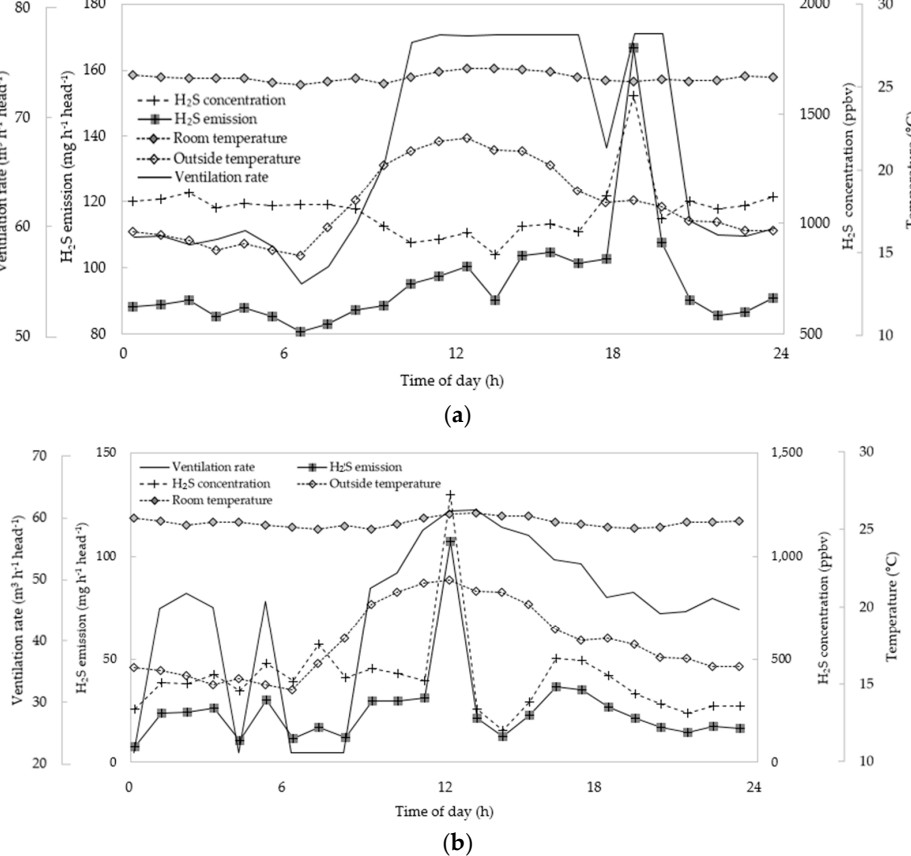

**Figure 8.** Diurnal variations of temperature, ventilation rate and generated $H_2S$ from (**a**) control (conventional slurry pit) room, (**b**) treatment (pit recharge system) room on day 5.

### 3.3. Gas Reduction Rates

The averages of daily mean $NH_3$, $H_2S$ concentrations and emissions are shown in Table 3. The average of each room temperature was about 25 °C, i.e., no statistical difference in control and treatment ($p > 0.05$). The reduction of gas concentrations was 37.1% and 79.8% for $NH_3$ and $H_2S$, respectively, for the room equipped with the pit recharge system. Reduction rates of emission were significant ($p < 0.0000$), i.e., 53.3% and 83.7% for $NH_3$ and $H_2S$, respectively.

**Table 3.** Average of daily mean $NH_3$ and $H_2S$ concentrations and emission rates in control (conventional slurry pit) and treatment (pit recharge system).

| | Control | Treatment | *p*-Value | Reduction Rate (%) |
|---|---|---|---|---|
| *n* | 14 | 13 | - | |
| Room temperature (°C) | 25.0 ± 0.7 [a] | 25.1 ± 0.6 [a] | 0.7125 | - |
| Ventilation rate ($m^3\ h^{-1}\ head^{-1}$) | 62.0 ± 11.7 | 47.0 ± 9.0 | - | - |
| Gas concentration | | | | |
| $NH_3$ (ppmv) | 14.9 ± 5.0 | 10.3 ± 3.8 | - | 32.6 ± 5.3 |
| $H_2S$ (ppbv) | 1,122 ± 137 | 239 ± 75 | - | 78.3 ± 6.8 |
| Gas emission rate | | | | |
| $NH_3$ ($g\ d^{-1}\ head^{-1}$) | 13.8 ± 4.5 [a] | 6.6 ± 2.4 [b] | 0.0000 | 53.3 ± 6.6 |
| $H_2S$ ($mg\ d^{-1}\ head^{-1}$) | 2,146 ± 311 [a] | 338 ± 92 [b] | 0.0000 | 83.7 ± 6.8 |

[a,b] Different superscripts in the same row meaning each group are significantly different ($p < 0.05$).

Due to the difference between average ventilation rates of control and treatment, we compared specially $NH_3$ concentrations and emission at similar ventilation rates (Table 4). In all three ventilation ranges, $NH_3$ concentration and emission from treatment were significantly lower than in control.

**Table 4.** Comparison of $NH_3$ concentrations and emissions from control and treatment rooms at similar ventilation rates.

| Ventilation Rate Range ($m^3\ h^{-1}\ head^{-1}$) | $NH_3$ Concentration (ppmv) | | | $NH_3$ Emission ($g\ h^{-1}\ head^{-1}$) | | |
|---|---|---|---|---|---|---|
| | Control (*n*) | Treatment (*n*) | Reduction Rate (%) | Control | Treatment | Reduction Rate (%) |
| 20~25 | 12.2 ± 8.4 [a] (40) | 7.6 ± 4.7 [b] (75) | 37.3 | 0.21 ± 0.16 [a] | 0.13 ± 0.08 [b] | 40.0 |
| 50~55 | 15.4 ± 5.8 [a] (33) | 6.6 ± 1.7 [b] (25) | 57.1 | 0.63 ± 0.24 [a] | 0.27 ± 0.07 [b] | 58.0 |
| 70~78 | 11.8 ± 3.4 [a] (138) | 8.8 ± 4.0 [b] (46) | 24.9 | 0.69 ± 0.20 [a] | 0.51 ± 0.23 [b] | 26.9 |

[a,b] Different superscripts in the same row meaning each group are significantly different. ($p < 0.05$).

In this study, normalized $NH_3$ and $H_2S$ emissions with 500 kg live weight (AU) were 41.5, 2.1 g per day, respectively. $NH_3$ emissions reported here were ~4.2 times higher than previous results (Table 1, Table A1). The estimated $H_2S$ emissions were higher than values reported by Lim et al. [35] and Kai et al. [34] and comparable with values reported by Blunden et al. [36], which used a commercial barn to estimate the gas emissions.

### 3.4. Characteristics of Recharging Liquid and Manure

Aerobically treated liquid manure collected on day 13 differed from the one collected on day 0 in several parameters. As shown in Table 5, electric conductivity (EC) decreased from 14.2 to 11.6 $\mu S\ cm^{-1}$. Although the total N contents were similar, most of the N was in the form of ammonium ($NH_4^+$) in day 0, while the one from day 13 had less ammonium nitrogen.

**Table 5.** Characteristics of recharged aerobically treated liquid manure (collected from the last stage of the aerobic treatment system, Figure 1). The samples were collected at the initial and final day of the experiment. Manure samples from the pit represent stored manure (Figure A2).

| Ventilation Rate Range ($m^3 \; h^{-1} \; head^{-1}$) | Aerobically Treated Liquid Manure | | Manure Sample from the Pit in Day 13 | | |
|---|---|---|---|---|---|
| | Day 0 | Day 13 | Control [1] | Treatment [2] | *p*-Value |
| Moisture content (%, w.b. [3]) | 98.7 | 98.7 | $93.9 \pm 2.8$ [a] | $95.6 \pm 4.0$ [a] | 0.0532 |
| Volatile solids (%, d. b. [4]) | 40.4 | 39.8 | $68.8 \pm 3.5$ [a] | $38.9 \pm 6.5$ [b] | 0.0000 |
| pH | 8.6 | 8.2 | $7.7 \pm 0.1$ [a] | $8.0 \pm 0.4$ [a] | 0.3475 |
| EC [5] ($\mu S \; cm^{-1}$) | 14.2 | 11.6 | $30.7 \pm 4.0$ [a] | $12.7 \pm 0.2$ [b] | 0.0159 |
| Total N ($mg \; L^{-1}$) | 1175.8 | 1199.9 | $6037 \pm 829$ | $2207 \pm 1173$ [b] | 0.0135 |
| $NH_4$-N ($mg \; L^{-1}$) | 918.2 | 216.3 | $3806 \pm 17$ [a] | $801 \pm 304$ [b] | 0.0002 |

[1] Manure from pit of control (conventional slurry pit); [2] Aerobically treated liquid manure mixed with manure from pit of treatment (pit recharge system); [3] wet basis; [4] dry basis; [5] Electric conductivity; [a,b] Different superscripts in the same row mean characteristics of manure sample from control and treatment pit are significantly different.

The manure characteristics of samples collected in control and treatment pits are compared in Table 5. A mixture of aerobically treated liquid manure mixed with stored manure had higher moisture content and lower volatile solids content compared with manure from control ($p < 0.05$). The mean total N content in samples from treatment was 2,207 mg $L^{-1}$, i.e., significantly lower than 6307 mg $L^{-1}$ in samples from control ($p < 0.05$). The $NH_4$-N content was 801 mg $L^{-1}$, i.e., significantly lower compared with 3806 mg $L^{-1}$ of the control ($p < 0.05$).

## 4. Discussion

### 4.1. Correlation of $NH_3$ & $H_2S$ Concentration with Ventilation Rate

In this study, the emissions of $NH_3$ and $H_2S$ were positively correlated with the ventilation rates in both rooms (Figures 3 and 7). However, in the case of gas concentrations, $NH_3$ showed no correlation with the ventilation rate in both rooms (R = $-0.09$ and 0.02, in control and treatment, respectively). Also, $H_2S$ showed a negative correlation with ventilation rates only in control (Figure 6). A more thorough analysis revealed that the properties of emitting manure near the surface in the treatment might explain these observations. Table 6 summarizes the detailed results about N content and the pH of manure sampled at three different depths of manure in each pit at day 13. The values from treatment showed a concentration gradient by sampled depth. The pH was higher, and the $NH_4$-N was lower in the (shallow) manure depth in treatment. The total N in shallow depth in control was 5414 mg $L^{-1}$, (over 2.5 times higher than in treatment, 2044 mg $L^{-1}$) and most of N existed in the form of $NH_4$. The $NH_4$-N in shallow depth in control was 3827 mg $L^{-1}$ and was over 6 times higher than treatment (619 mg $L^{-1}$). Taken together, the manure surface in both control and treatment (represented by shallow depth data) had plenty of total N and $NH_4$-N to facilitate emissions. The $NH_4$-N is sensitive to temperature and pH and can be easily converted into $NH_3$ gas [40]. If the available N was identical in the control and treatment, a higher pH would result in higher emissions from treatment. However, the opposite effect was observed. A significant difference in $NH_3$ emissions and a mitigation effect occurred because of the significant differences in the total N and $NH_4$-N in control and treatment. This was likely the reason that the ventilation rate increase did not result in a dilution effect for $NH_3$ concentration while other factors (e.g., pH, $NH_4$-N and the total N) were offsetting any potential effect.

In contrast, slightly basic (pH 7.7) manure in shallow depth of control prevented the sulfur volatilization, and the emitted $H_2S$ was diluted by increasing ventilation rates (Figure 6). Approximately 80% of $H_2S$ converted to HS- when the pH is 7.7 at 25 °C. However, in treatment (pH 8.3), there was no correlation between $H_2S$ concentration and ventilation rates. It appears that aerobically treated liquid manure in the pit was at least in part responsible for the significant reduction of $H_2S$ emissions. The more basic pH (8.3) of manure in shallow depth was also helpful to reduce $H_2S$ emissions (Tables 6 and 7).

**Table 6.** Characteristics of manure collected at 3 different depths (shallow, middle, deep) of each pit (Figure A2).

| | Control [1] | | | | Treatment [2] | | | |
|---|---|---|---|---|---|---|---|---|
| | Mean | Shallow [3] | Middle [4] | Deep [5] | Mean | Shallow [3] | Middle [4] | Deep [5] |
| Moisture content (%, w.b. [6]) | 93.9 | 96.3 | 95.2 | 90.1 | 98.5 | 98.6 | 98.3 | 89.9 |
| Volatile solids (%, d. b. [7]) | 68.8 | 66.5 | 66.2 | 73.8 | 43.2 | 40.8 | 45.7 | 30.2 |
| pH | 7.7 | 7.7 | 7.7 | 7.6 | 8.0 | 8.3 | 8.2 | 7.5 |
| EC [8] ($\mu S\ cm^{-1}$) | 30.7 | 33.2 | 32.8 | 26.1 | 12.7 | 12.9 | 12.7 | 12.6 |
| Total N ($mg\ L^{-1}$) | 6037 | 5414 | 5,488 | 7209 | 2,207 | 2,044 | 1125 | 3453 |
| $NH_4$-N ($mg\ L^{-1}$) | 3806 | 3827 | 3,785 | 3807 | 801 | 619 | 633 | 1153 |

[1] Manure from the pit of control (conventional slurry pit); [2] Aerobically treated liquid manure mixed with manure from the pit of treatment (pit recharge system); [3] 60 cm from the bottom; [4] 40 cm from the bottom; [5] 20 cm from the bottom [6] wet basis; [7] dry basis; [8] Electric conductivity.

**Table 7.** Percent of $H_2S$ and $HS^-$ ($H_2S/HS^-$ equilibrium) in different pH of liquid at 25 °C.

| pH | Fraction Present (%) | |
|---|---|---|
| | $H_2S$ | $HS^-$ |
| 7.7 (Control) | 17.95 | 82.05 |
| 8.3 (Treatment) | 5.21 | 94.79 |

## 4.2. NH₃ & H₂S Concentration and Emission Rates in the Pit Recharge System

The pit recharge system showed a significant reduction in $NH_3$ (32.6%) and $H_2S$ (78.3%) concentrations. That reduction of gas concentrations is beneficial to improve indoor air quality and working environment for farmers. Further research in the effects of improved air quality on the productivity of pigs is warranted. Although the reduction rate of $NH_3$ and $H_2S$ emissions by pit recharge system were 47.8% and 82.3%, respectively, the actual gas emissions in this study were greater than previous studies with pit recharge system (Table A2). We hypothesize that one of the reasons was due to the 17.5% crude protein content in the feed, which was greater than generally recommended in feeding standard in other countries (Table 8). Generally, ~20% more $NH_3$ is produced per 1% point of crude protein increase [41]. Based on the simple calculation, 31–124% more $NH_3$ would be produced compared with that reported by Lim et al. [35] and Kai et al. [34], due to the 1.5–6%-point higher crude protein. This is consistent with measured $NH_3$ concentrations (Table A2).

**Table 8.** Comparison of crude protein contents in feed and previously recommended crude protein contents by feeding standard.

| | Crude Protein in the Feed (%, d.b. [1]) |
|---|---|
| JRC (2005) [1] | 13.0 |
| SCA (1987) [2] | 15.0 |
| Lim et al. (2004) [35] | 11.5~13.1 |
| Kai et al. (2006) [34] | 16.0 |
| This study | 17.5 |

[1] Agriculture, Forestry and Fisheries Research Council Secretariat of Japan; [2] Standing Committee on Agriculture in Australia.

## 4.3. Characteristics of Recharging Liquid and Manure

During 14 days of the experiment, the changes of N species (forms) in aerobically treated liquid manure was observed (Table 5). The total N contents of day 0 and day 13 were very similar, but most of N was in the form of $NH_4$ on day 0 while the $NH_4$-N concentration was low at day 13. During the aerobic treatment, the organic N in manure converts to $NH_4$ by mineralization. If more aeration is available, the $NH_4$ converts to $NO_3$ (a stable N form by nitrification) and pH decreases. Therefore,

it is expected that it will be possible to control the $NH_3$ and $H_2S$ emission rates by controlling the nitrification level of the liquid manure by (e.g.,) aeration rate in the aerobic treatment system.

It is also important to consider the mass balance of N and S in a pit recharge system. This paper reported a significant improvement to air quality (i.e., lower gas concentrations) inside the barn and significantly lower emissions from the barn. However, it is reasonable to expect $NH_3$ and $H_2S$ emissions from liquid-solids separation, and liquid manure aerobic treatment system. Comparing the total N contents of aerobically treated liquid manure and manure from the control pit shows that about 80% of N was lost during the aerobic treatment of swine manure (Table 5).

The centralized manure plants, e.g., like the one that treats manure from conventional storage at this tested farm, have mandatory deodorization systems for treating emissions. For example, the two stages of liquid manure aeration areas are covered, and odorous air is treated by a wet scrubber before emitted to the atmosphere. Also, odorous air from solid-liquid separation and solid composting are also treated by the same wet scrubber. Thus, the technology to treat gaseous emissions on a large centralized manure plant scale is in place. However, the economic feasibility of installing such treatment on a farm scale needs to be evaluated.

## 5. Conclusions

The evaluation of a semi-continuous pit recharge system on the mitigation of ammonia ($NH_3$) and hydrogen sulfide ($H_2S$) emissions from a swine finisher barn was conducted for two weeks at a commercial swine farm in the Republic of Korea. Gas concentrations and emissions from a room equipped with a pit recharge system were compared with a room operating conventional slurry pit under the slatted floor. Mean reduction of $NH_3$ and $H_2S$ emissions were $49 \pm 6\%$ ($p = 0.0001$) and $82 \pm 7\%$ ($p < 0.0000$), respectively. The removal efficiency of $H_2S$ was higher than $NH_3$ likely because the pH of aerobically treated liquid manure remained slightly above 8. It is also expected that it will be possible to control the $NH_3$ and $H_2S$ removal rates by controlling the nitrification level of the liquid manure in the aerobic treatment system. More work is warranted to assess the N balance in this system, and emissions of odor and GHGs. The economic feasibility of installing and maintaining such treatment on a farm scale and possible effects on animal productivity needs to be evaluated.

**Author Contributions:** Conceptualization, H.A.; methodology, H.A.; validation, J.W., J.A.K. and H.A.; formal analysis, J.W.; investigation, J.W., S.L., E.K., M.L.; resources, S.L., E.K., M.L. and H.A.; data curation, J.W.; writing—original draft preparation, J.W.; writing—review and editing, J.W., J.A.K., and H.A.; visualization, J.W.; supervision, H.A. and J.A.K.; project administration, H.A.; funding acquisition, H.A.

**Funding:** This research was supported by Korea Institute of Planning and Evaluation for Technology in Food, Agriculture, Forestry and Fisheries (IPET) through Agri-Bio Industry Technology Development Program, funded by Ministry of Agriculture, Food and Rural Affairs (MAFRA) (Grant No. 317008-3). This project was partially supported by the Iowa Agriculture and Home Economics Experiment Station, Ames, Iowa. Project No. IOW05556 (Future Challenges in Animal Production Systems: Seeking Solutions through Focused Facilitation) is sponsored by Hatch Act and State of Iowa funds.

**Acknowledgments:** This authors gratefully acknowledge Seongyoong Cho (Nagwon Chuksan, Swine Farm) and Jongkook Lee (EcoViron Company) for providing experimental swine farm and pit recharge system. Special thanks to Woosang Lee (Smart Control & Sensing Inc.) for his help with gas monitoring.

**Conflicts of Interest:** The authors declare no conflict of interest. The funders had no role in the design of the study; in the collection, analyses, or interpretation of data; in the writing of the manuscript, or in the decision to publish the results.

## Appendix A

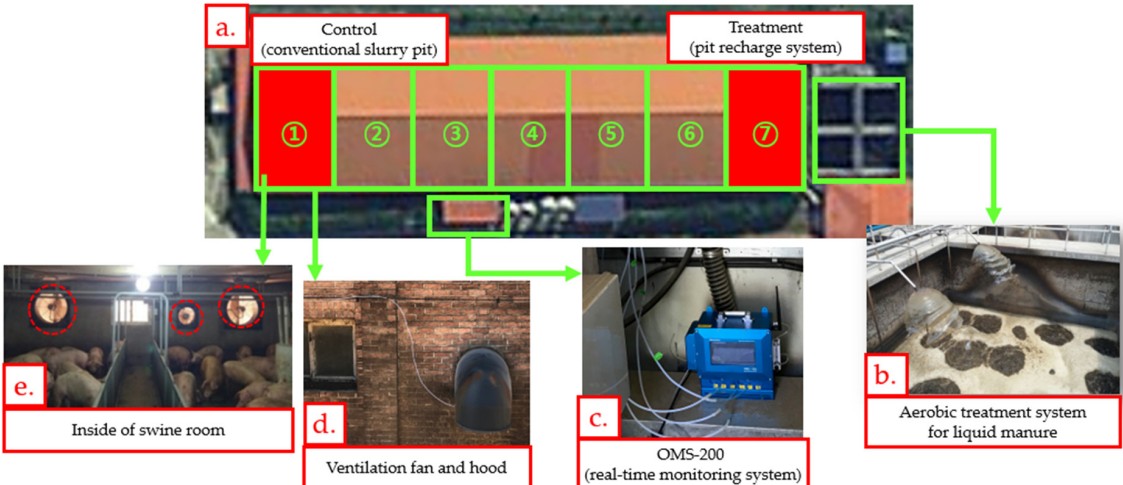

**Figure A1.** Pictures of experimental site. From top clockwise: a. Satellite photo swine barn with marked control & treatment rooms (rooms 1 to 3 operate conventional slurry pits, 5 to 7 operate pit recharge, 4 could be used in both modes), b. Location of the outdoor aerobic treatment system for liquid manure; view of manure spray nozzles discharging into the liquid manure treatment system for minimizing foam generation from aeration, c. Real-Time gas and ventilation rate monitor, d. Primary fan with the downward pointed cone, air sampling line visible to the left of fan cone, e. interior of swine barn.

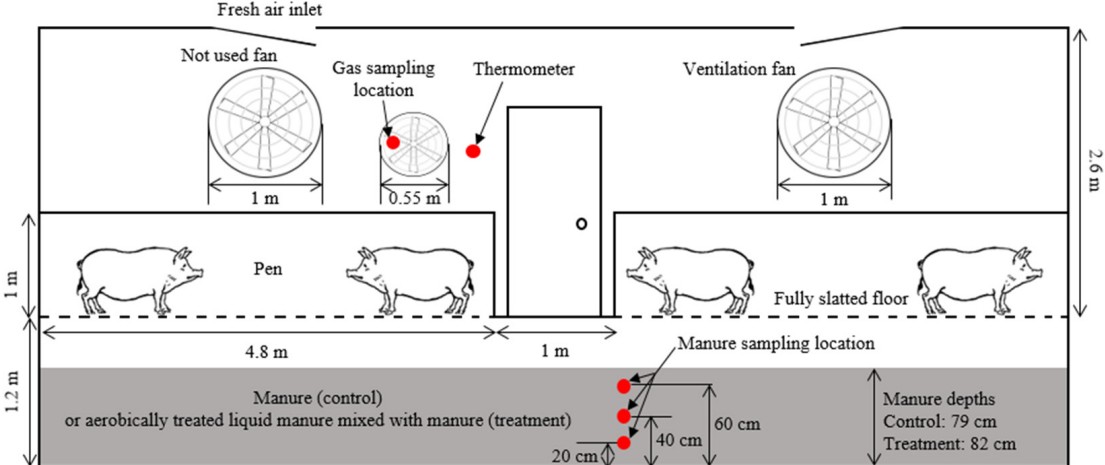

**Figure A2.** Side view of tested swine rooms (control and treatment). The gas sampling port was positioned immediately downstream from a primary fan (operating at all-times). The thermometer was installed near the fan. The manure was sampled from the middle of the pit.

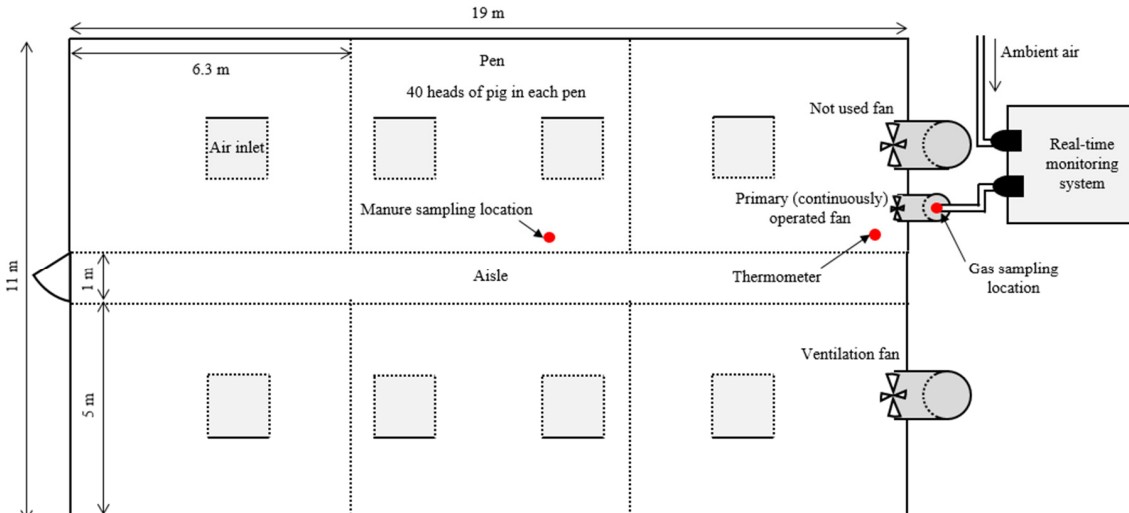

**Figure A3.** Top view of tested swine room.

**Table A1.** Specifications of gas sensors (Membrapor Co.) used in this study.

|  | $NH_3$ | $H_2S$ |
|---|---|---|
| Model | NH3/CR-50 | H2S/C-50 |
| Detecting range | 0–50 ppm | 0–50 ppm |
| Resolution | 0.5 ppm | 50 ppb |
| Linearity ($R^2$) | 0.9951 | 0.9995 |

**Table A2.** Comparison of $NH_3$ and $H_2S$ emissions with previous literature.

| Study | Recharging Liquid | Recharging Frequency | Gas Emission Rate ($g\ d^{-1}\ AU^{-1}$) |
|---|---|---|---|
| Lim et al. (2004) [35] | Lagoon effluent | 1 wk | $NH_3$; 10<br>$H_2S$; 0.16 |
|  |  | 2 wks | $NH_3$; 12<br>$H_2S$; 0.34 |
|  |  | 6 wks | $NH_3$; 11<br>$H_2S$; 1.42 |
| Kai et al. (2006) [34] | Water | 1 wk | $NH_3$; 17~23 |
| Blunden et al. (2008) [36] | Lagoon effluent | 1 wk | $NH_3$; 40.8<br>$H_2S$; 4.2 |
|  |  | 1 wk | $NH_3$; 37.1<br>$H_2S$; 3.3 |
|  |  | 1 wk | $NH_3$; 29.5<br>$H_2S$; 1.2 |
|  |  | 1 wk | $NH_3$; 14.3<br>$H_2S$; 1.7 |
| This study | Aerobically treated liquid manure | 3 times $d^{-1}$ | $NH_3$; 41.5<br>$H_2S$; 2.1 |

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
