# Peer review of "Evaluation of Semi-Continuous Pit Manure Recharge System Performance on Mitigation of Ammonia and Hydrogen Sulfide Emissions from a Swine Finishing Barn"

_atmosphere, doi:10.3390/atmos10040170_

Reviewer 1 Report

1. Introduction

P2 L17: Sentence starting with "The source-based……" seem not to be leading anywhere. Please consider rephrasing.

P2 L21-22: Reference [21] does not support the sentence.

P0?? L20-21: I disagree. Maybe due to national legislation in Korea, but from an environmental point of view, aerobic treatment of manure reduces the amount of N, thus reducing the fertilizer value.

2. Experiments

P0? L38: I think "which" should be replaced with "with".

P0 L49: add unit (m) to the figure describing the depth of the manure pits.

P1 L1: the sentence starting with "i.e., there was no ….." is not clear to me. I think what is meant is "i.e., there was no air exchange between rooms". Please clarify and consider rephrasing.

P1 L5: "automatically" does not indicate the mode of operation of the fans. Were the to fans variable speed fans? Please clarify and rephrase. 

P1 L9: what does the asterisk mean?

Figure 2: It remains unclear to me whether the amount of manure increased in the treatment pits or if the manure level was constant.

P2 L4-10: To my knowledge, electrochemical sensors are mainly developed and used as "alarm sensors". The results show that especially the observed H2S concentrations are very low. Thus, performance data on the sensors (detection limit, linearity etc.) should be incorporated in the manuscript.

P2 L11: The measuring principle of the AMA is unclear to me. Please clarify.

Equation (1): according to the equation no correction for ambient air is carried out. Nearby emission sources such as neighboring pig rooms may significantly affect gas concentrations in ambient air entering the rooms. This must be accounted for, or at least it must be shown that ambient air did not contain NH3 and H2S in significant amounts.

P3 L2: Defining a method as "the total N method" is not specific enough. Please clarify.

Figures 4, 5, 8, and 9: why are there no data for ventilation rates between 22 and 40 m3/h per head? I assume this has to do with the operation of the ventilation system with three fans. Please consider elaborating on this.

P3 L6: Is the T-test an appropriate statistical method to use considering the test design ? This is in fact a time series. Data is not independent which is a prerequisite for the T-test. Please consider a more advanced method.

Table 3: The ventilation rate in the treatment room was significantly lower than in the control room (47 vs. 62 m3/h per head). Why is that? The treatment can hardly be responsible for such a difference?! Since you find a correlation between ventilation rate and emission rate, this can be considered a confounding factor. Using the equation in figure 5 it can be shown that if the ventilation rate in the treatment room was the equal to the control room, the average emission in the treatment room would be estimated 12.1 g/d per head instead of 9 or 29 % lower than the control room. Still a reduction but quite a bit lower compared with 47.8% as indicated in the table. I believe the same applies to H2S. Please elaborate on this.

Table 5: You measured 896 mg NO3- in untreated pig manure. How is this possible? There is almost no oxygen content in untreated pig manure and thus no nitrification. Please substantiate eg. via references.

Looking at the figures in table 5, it occurs to me that the aerobic treatment apparently was not "in balance". The NH4+ decreased dramatically from day 0 to day 13 whilst NO3- increases. Or is it a matter of "natural" variation? In any case this may influence the emission rate of NH3. Therefore, please clarify and elaborate.

P11 L6: "Approximately about" is redundant.

P11 L13: Suggests to add "The" to begin the sentence "Pit recharge…".

P11 L19: Suggests to add "was" to "which (was) greater….."

P12 10-15: The authors correctly mention the importance of N balance and points to the apparent N loss of approx 80 % during the aerobic treatment, but indicate that the aerobic treatment plants are equipped with a cover and a wet scrubber. This however is still problematic from an environmental point of view. How much is lost as N2O? This is a powerful greenhouse gas known to be produced in such system as well as in biological wet scrubbers. Please elaborate on this.

Author Response

We appreciate the opportunity to address all comments (see attached). 

Reviewer 2 Report

Very interesting study comparing two swine production systems (pit-type versus aerobic treatment-recharge).  The authors are primarily interested the effects of the recharge system on gas emissions and air quality within the production house.  There are a few awkwardly written sentences and possibly a few unnecessary figures/tables that need to be considered. 

I have two issues that need to be addressed: 

First, I was a bit confused by the volume of material in the control and recharge rooms.  Can you please add information about the volumes--maybe the difference in concentrations is just related to the difference in liquid?  Was there always 12 cubic meters of volume in both pits?  I think from the figures, that is correct, but it is not clear.

Second, ventilation rate may be the primary cause for higher NH3 emission since the control room consistently had much higher ventilation rates during the study (figure 4).  Are you stripping out NH3?  True, the slurry concentration is lower in the recharge room.  Maybe consider figure 5--at similar ventilation rates, the emission might be slightly lower in the recharge room, but is it significant?  Look at figure 9--for H2S at any ventilation rate, there is a clear difference.  For H2S, you don't have this issue.  I think it is appropriate to compare the daily differences in concentration and flux, but it is important to try and make comparisons for similar ventilation rates.

Minor comments:

P1 L15: "In this research, for the first time," sounds awkward.  How about "A semi-continuous pit manure recharge system was evaluated for its capacity to mitigate..."

P1 L19: fraction--after reading the whole paper, was this a complete pump out, or was only a fraction removed and replaced?  General percentage pumped would be good.

P1 L33, L34: No 'the' needed

P1 L41: Interaction of odorous VOCs

P1 L42: Remove 'Relatively few'.  Sounds awkward and like you are stretching to make a case for your work.

P2 L32-39:  Duplicates what is in the table.  Just summarize with "varying effectiveness".

P4 L12:  I think you mean slurry and not slutty!

P4 L15: See P1 L15.

P4 L18-24:  Just speak to the general use of the system.  What is it developed to address?  Delete 'tried as'

Fig2 and Fig3--I don't think these are necessary.  Information in the text.

P7 Stats:  Need a bit more information--did you do a paired test for data on the same date/time? 

Fig 4 and 7--ventilation rate confounds the effect of treatment when you look just at time.  You really need to compare emissions at similar fluxes (and maybe control for a 48 hour period so you don't compare day 1 to day 14).

P11 L9: Use 18:00 instead of 18 o'clock

Table 4 is not necessary since 95% of it is in table 1.  Just report the 'this study' data in the text and refer to table 1.

Table 5:  Is there a volume difference?  Mass comparison would be more effective.  WHAT ABOUT DISSOLVED H2S?  Too bad if you didn't collect that data...

Author Response

We appreciate the opportunity to address all comments (attached).
